# Enhancing the Dye-Rejection Efficiencies and Stability of Graphene Oxide-Based Nanofiltration Membranes via Divalent Cation Intercalation and Mild Reduction

**DOI:** 10.3390/membranes12040402

**Published:** 2022-04-02

**Authors:** Hobin Jee, Jaewon Jang, Yesol Kang, Tasnim Eisa, Kyu-Jung Chae, In S. Kim, Euntae Yang

**Affiliations:** 1Department of Marine Environmental Engineering, Gyeongsang National University, Tongyoung 53064, Korea; hbj99@gnu.ac.kr; 2KEPCO Research Institute (KEPRI), Korea Electric Power Corporation (KEPCO), Naju 58277, Korea; jwjang@gm.gist.ac.kr; 3School of Earth Sciences and Environmental Engineering, Gwangju Institute of Science and Technology (GIST), Gwangju 61005, Korea; yesol7964@gist.ac.kr (Y.K.); iskim@gist.ac.kr (I.S.K.); 4Department of Environmental Engineering, Korea Maritime and Ocean University, Busan 49112, Korea; tasnim.i.eisa@gmail.com (T.E.); kkjdream@kmou.ac.kr (K.-J.C.)

**Keywords:** divalent ion, graphene oxide, membrane, nanofiltration, reduction, crosslinking

## Abstract

Laminar graphene oxide (GO) membranes have demonstrated great potential as next-generation water-treatment membranes because of their outstanding performance and physicochemical properties. However, solute rejection and stability deterioration in aqueous solutions, which are caused by enlarged nanochannels due to hydration and swelling, are regarded as serious issues in the use of GO membranes. In this study, we attempt to use the crosslinking of divalent cations to improve resistance against swelling in partially reduced GO membranes. The partially reduced GO membranes intercalated by divalent cations (i.e., Mg^2+^) exhibited improved dye-rejection efficiencies of up to 98.40%, 98.88%, and 86.41% for methyl orange, methylene blue, and rhodamine B, respectively. In addition, it was confirmed that divalent cation crosslinking and partial reduction could strengthen mechanical stability during testing under harsh aqueous conditions (i.e., strong sonication).

## 1. Introduction

Water shortage is currently one of the most critical global issues that needs to be resolved [1]. To address the challenge of water shortage, technologies for obtaining clean water, such as those based on water reuse and desalination, have been gaining significant attention [2,3,4]. Among these processes for securing water from alternative sources, nanofiltration (NF) has increasingly been employed because it has several attractive merits, such as the capability to reject ions up to the divalent level (e.g., heavy metal ions), the ability to remove the widest range of organic contaminants, and a relatively low energy consumption [5,6,7,8,9,10]. However, despite their increasing popularity, NF processes also have several drawbacks that need to be addressed. In particular, membrane-related issues, such as irreversible fouling and a tradeoff between solute rejection and water permeability, are often considered the most critical problems that need to be overcome before NF processes become commonly used for water treatment.

These membrane-related issues are mainly derived from the intrinsic properties of polymeric membranes, which are the most commonly used for NF; therefore, various efforts have been made in the field of membrane sciences to discover novel building blocks to substitute conventional polymeric membranes for next-generation NF membranes [11,12]. Recently, graphene oxide (GO), which is a carbon-based 2D nanomaterial decorated with various oxygen functional groups, has been investigated as a possible new high-performance membrane material for its outstanding physical and chemical characteristics, such as its excellent mechanical and chemical stability, ultrathin 2D structure, and high hydrophilicity [13,14]. With the laminar stacking of GO nanosheets, thin-film laminates with a unique structure consisting of hydrophobic *sp*^2^ regions and hydrophilic *sp*^3^ regions can be created [15]. This unique laminar structure can provide nanochannel (i.e., interlayer spacing) galleries with an almost frictionless property that allows for the abnormally rapid transport of water molecules to the continuously hydrophobic *sp*^2^ region [16,17]. These distinctive nanochannels are also size-controllable, resulting in a tunable and precise molecular sieve [18,19,20,21]. Moreover, the oxygen functional groups of GO can grant hydrophilic properties to the surface of the GO membranes (GOMs), as well as make it easier to tune their characteristics [22,23].

However, to commercialize GOMs and, furthermore, to make them an alternative membrane platform to the current conventional polymeric membranes, improvements need to be made in terms of their stability and low solute-rejection efficiencies, which are caused by the GO stacking loosening when the GO nanosheets are hydrated [24]. Furthermore, for targeted NF applications, the size of the nanochannels created between adjacent GO nanosheets needs to be controlled at the angstrom level.

Recently, among the strategies employed to resolve these issues for GO-based NF membranes were crosslinking between a laminar stacking of GO nanosheets with various cations and eliminating oxygen functional groups in GO nanochannels [19,21,25,26,27]. For example, Park et al. (2008) [27] improved the mechanical stiffness and fracture strength of GO films by up to 200% and 50%, respectively, by crosslinking neighboring GO nanosheets with divalent cations (i.e., Mg^2+^ and Ca^2+^). They suggested that binding cations with oxygen functional groups to the GO edges and to the graphitic basal plane of the GO nanosheets could enhance the mechanical strength of the GO films. Chen et al. (2017) [28] tuned the size of the GO nanochannels to a range from 11.4 Å to 13.6 Å by inserting various ions, such as Na^+^, K^+^, Li^+^, Mg^2+^, and Ca^2+^, into the GO laminates. The GOMs tuned to these nanochannel sizes with these cations had the ability to exclude larger ions than those tuned with the intercalated cation. Yu et al. (2017) [29] also evaluated various cation-modified GO laminates. In their study, GO laminates crosslinked with trivalent ions, such as Al^3+^, exhibited high stability, even in a NaCl solution, as well as superior water flux, compared with those crosslinked with other cations (i.e., Ca^2+^, Mg^2+^, and Na^+^). Similarly, Liu et al. (2017) employed trivalent ions, such as Al^3+^ and Fe^3+^, as crosslinkers to fabricate robust GO-based membranes for natural organic matter removal. The trivalent crosslinked GOMs, particularly GOMs crosslinked with Fe^3+^, showed excellent performance in terms of water flux, organic contaminant removal, fouling resistance, and stability compared with pristine GOMs. It was also found that the interlayer spacing changed with different ion concentrations. Yang et al. (2018) [19] and Qing Zhang et al. (2018) [30] demonstrated that the water permeability and solute-rejection efficiencies of GOMs were tailorable by adjusting the nanochannel size of GOMs, specifically by controlling the degree of reduction in the laminar GO layers. More recently, Yuan et al. (2021) [16] developed highly stable GOMs with a high NaCl rejection efficiency of 91% by intercalating K^+^ ions into the rGO layers.

As seen above, previous work demonstrated that crosslinking stacked GO layers with cations and controlling the degree of reduction in stacked GO layers serve as versatile approaches to concurrently tuning the nanochannel size of GOMs at the angstrom level and using them to enhance the stability in aqueous solutions. In addition, the cation crosslinking and reduction can be achieved with simple, cost-effective steps. Therefore, cation crosslinking and partial reduction are reasonable strategies for fabricating robust GO-based NF membranes with a customized separation capability.

Therefore, to develop high-performance NF membranes that have an efficient removal rate involving small-size organic dye molecules (300–500 g/mol), we crosslinked GOMs with Mg^2+^ ions and then partially reduced the Mg^2+^-crosslinked GOMs with heat treatment, considering the following hypotheses: (1) Mg^2+^ ions can create a relatively large interlayer spacing [28] that can be expected to secure moderately loose nanochannels, which would allow for ultrafast water permeation but would efficiently exclude targeted organic dyes, and (2) partial reduction may help maintain the GO nanochannel size and enhance the stability of GOMs by preventing excessive swelling of the laminar GO layer.

## 2. Materials and Methods

### 2.1. Materials

A highly concentrated GO solution (6.2 g/L, Graphene Laboratories Inc., New York, NY, USA) was diluted in deionized water to 6.0 mg/L. A nylon membrane filter (C_10_H_20_(CO)_2_(NH)_2_, 47 mm, pore size: 0.20 μm) was obtained from the CHMLAB Group, Spain. Magnesium chloride (MgCl_2_, >97.0% in purity, Junsei Chemical Co., Ltd., Tokyo, Japan) was dissolved in deionized water to 0.2 M. Methyl orange (MO, C_14_H_14_N_3_NaO_3_S, extra pure, Junsei Chemical Co., Ltd., Tokyo, Japan), methylene blue (MB, C_16_H_18_ClN_3_S, >95% in purity, Junsei Chemical Co., Ltd., Tokyo, Japan), and rhodamine B (RB, C_28_H_31_ClN_2_O_3_, extra pure, Daejung chemicals & materials Co., Ltd., Siheung-si, South Korea) were dissolved in deionized water to 10 ppm.

### 2.2. Membrane Fabrication

First, the GO solution (6.0 mg/L) of 50 mL was filtrated through a nylon membrane using a lab-scale dead-end filtration system to prepare a pristine GOM, as described in Figure 1. After that, the collected GOM was dried in a vacuum desiccator for 12 h at room temperature. To intercalate the Mg^2+^ ions, the dried GOM was soaked in a 0.2 M MgCl_2_ solution at room temperature for 24 h. Then, it was rinsed in deionized water, followed by drying for 12 h. The Mg^2+^-intercalated GOM (Mg-GOM) was reduced by heat treatment at a temperature of 150 °C in a vacuum for 30 min. The GO membrane prepared by thermal reduction is referred to as Mg-rGOM(TH). All of the prepared membranes were stored in a desiccator before use.

### 2.3. Membrane Characterization

The morphological structures of the surface and the cross-section of the prepared GOMs were observed with a field-emission scanning electron microscope (FE-SEM, TESCAN, Mira, Brno, Czech). The surface hydrophilicity and surface charge of the GOMs were measured using a contact angle analyzer (SEO, phoenix-300, Suwon, South Korea) and a zeta potential analyzer (ELSZ-2000, Otsuka, Japan), respectively. The chemical structure and atomic composition were analyzed using Fourier-transform infrared spectroscopy (FT-IR, Thermo fisher, IS50, Waltham, MA, USA), X-ray photoelectron spectroscopy (XPS, Thermo Fisher Scientific, Nexsa G2, Waltham, MA, USA), and SEM with energy-dispersive X-ray spectroscopy (SEM-EDX, Hitachi, S4700, Tokyo, Japan). The crystallinity and sizes of the nanochannels of the GOMs were determined using an X-ray diffractometer (XRD, Rigaku SmartLab, Tokyo, Japan).

### 2.4. Nanofiltration Performance Evaluation

To evaluate the performance of the GOMs, the pure water permeability and solute-rejection rates were estimated during the operation of a custom-made dead-end filtration system with an effective membrane area of 12.25 cm^2^ under an applied pressure of 1 bar for 5 h after 1 h stabilization. The weight data of the permeates were automatically collected using an electronic balance (Ohaus, PAG4102C, Parsippany, NJ, USA) with data acquisition software (SPDC data collection) at intervals of 1 min. The pure-water permeabilities of the GOMs were calculated using the acquired weight data according to the following equation:(1)$$J=\frac{V}{A\times t\times P},$$
where J is the pure-water permeability, A is the effective area of the tested membranes (12.25 ${\mathrm{cm}}^{2}$), t is the time, P is the applied pressure (1 bar), and V is the volume of permeate.

To evaluate the solute rejection of the GOMs, 10 mg/L each of three different organic dye solutions (i.e., methyl orange (MO), methylene blue (MB), and rhodamine B (RB)) was used. The concentration of organic dye solutions was determined using an ultraviolet–visible spectrophotometer (UV–Vis, HS-3700, Humas, Daejeon, South Korea). The organic dye-rejection efficiencies were computed using the following equation:(2)$$R=\left(1-\frac{C_{p}}{C_{f}}\right)\times 100\%,$$
where R is the rejection rate, and $C_{f}$ and $C_{p}$ are the feed and permeate concentrations, respectively.

### 2.5. Membrane Stability Assessment

The mechanical stability of the prepared membranes under aqueous conditions was assessed in a sonication bath (40 kHz, Power sonic 510, HwaShin Tech., Gwangju, South Korea). To this end, the dried GOMs were first fixed on the bottom of Petri dishes. Then, the Petri dishes were filled with deionized water. Finally, the Petri dishes were placed in a sonication bath, and any physical changes in the GOMs were observed with the naked eye for 90 min.

## 3. Results and Discussion

### 3.1. Morphological Properties of GOM Series

The prepared GOMs showed a clear difference in color through optical observation, as shown in Figure 2A. After soaking the GOM in a MgCl_2_ solution, the surface color of the GOM changed from a glossy gold to a matte brown. This can be attributed to the Mg^2+^ ions of the oxygen functional groups occupying the GOM surface. In a previous study, it was observed that the adhesion of metal ions with oxygen-containing groups to GO nanosheets led to a decrease in intensity and a shift in position of the peak generated by the oxygen functional groups of GO nanosheets during a UV–Vis spectrum measurement of GO dispersion [31]. Again, after thermal treatment of Mg-GOM, the surface became slightly metallic and darker. This was because visible light could be absorbed by the graphitic *sp*^2^ bonds in the GO nanosheets recovered during thermal reduction of the Mg-GOM [32].

However, during the top and cross-sectional morphology analyses using SEM, there was no evident difference observed. As shown in Figure 2B, nano-wrinkle structures existed on the surfaces of all GOM series. In addition, the cross-sectional views show that all types of prepared GOMs had laminar structures, and their thickness was estimated to be about 150 nm (Figure 2C).

### 3.2. Effect of Mg^2+^ Crosslinking on the Chemical Structure of GOM Series

For a more accurate observation of the effect of the intercalation of Mg^2+^ ions and partial reduction on the prepared GOMs, the chemical composition and structure were analyzed with EDX, XPS, and FT-IR. First, in the analysis of changes in the chemical status of the GOMs after the intercalation of Mg^2+^ ions, as presented in Figure 3, no peak for magnesium (Mg) was seen during the EDX analysis of the GOM, whereas an obvious Mg peak appeared during the EDX analyses of the Mg-GOM. Moreover, EDX mapping demonstrated a homogeneous distribution of Mg elements in the Mg-GOM (Appendix A). This EDX result is analogous to the result obtained from the XPS analysis. The GOM showed no peak for Mg in the XPS spectrum, but the Mg-GOM exhibited a prominent Mg peak (Figure 4A). The EDX and XPS results confirm the existence of Mg^2+^ ions in the GOM after immersing the GOM in an MgCl_2_ solution.

In addition, the variations in the C 1*s* XPS spectra of the GOM series, indicating bonds related to the C and O atomic elements, were found after intercalation with the Mg^2+^ ions, as displayed in Figure 4B. The C 1*s* spectra can be split into three types of carbon bonds, i.e., C=C, C–O, and O–C=O. First, for the GOM, peaks for each bond appeared at 284.78 eV for C=C, 286.78 eV for C–O, and 288.28 eV for O–C=O. However, for the Mg-GOM, the positions of the C=C and O–C=O peaks were slightly shifted to 284.88 eV and 288.48 eV, respectively. In terms of variations in the peak intensity, the Mg-GOM exhibited an increased peak intensity for the C=C bond but a decreased peak intensity for the C–O bond and the O–C=O bond compared with the GOM. The changes in the peak intensities resulted in changes in the atomic content ratio of the three split C-related bonds after the intercalation of Mg^2+^ ions, as shown in Figure 4C. The variations in XPS spectra between the GOM and the Mg-GOM can be attributed to several reasons. Firstly, the ring structures of the epoxy-functional groups of GO nanosheets were opened due to the formation of coordinate covalent bonds between the Mg^2+^ ions and the epoxy groups [25,27]. Secondly, Mg^2+^ ions could create alkoxides by combining the hydroxyl and carbonyl groups of the GO nanosheets. Lastly, Mg^2+^ ions could form noncovalent interactions with the graphitic basal plane regions of the GO nanosheets.

These results can also be corroborated by the results of the FT-IR analysis (Figure 4D). From the FT-IR spectrum of the GOM, a broad peak between 3000 cm^−1^ and 3500 cm^−1^ for OH stretching [19], a peak at around 1719 cm^−1^ for C=O stretching [33], a peak at around 1581 cm^−1^ for the C=C bond [34], a peak at around 1364 cm^−1^ for the C–O vibrational band, a peak at around 1221 cm^−1^ for epoxy C–O stretching vibration, and a peak at around 1041 cm^−1^ for alkoxy C–O stretching were found, which are the commonly found peaks for GO nanosheets [25]. However, the peak at around 1719 cm^−1^, indicating C=O stretching, and the peak at around 1221 cm^−1^, representing epoxy C–O stretching vibration, were diminished in the spectrum of the Mg-GOM. Moreover, a decrease in peak intensity for alkoxy C–O stretching (1041 cm^−1^) was observed in the spectrum of the Mg-GOM. These findings suggest the formation of ring-opened epoxide groups, metal–carboxylate interactions, and noncovalent cation–π interactions [25]. Through a collective analysis of the above results and due to the widened interlayer spacing of the Mg-GOM (the details are mentioned later), it is believed that the layered GO nanosheets were successfully crosslinked with Mg^2+^ ions in the membranes.

### 3.3. Effect of Thermal Reduction on the Chemical Structure of GOM Series

The effect of thermally reducing the divalent cation-intercalated GO layers on the chemical status of the GOM was also thoroughly investigated. First, the presence of Mg^2+^ ions in the membrane was confirmed with the observation of an Mg peak in both the EDX and XPS patterns of the Mg-rGOM(TH) (Figure 3 and Figure 4A). Additionally, there were no significant changes in the atomic percentage of Mg after thermal reduction (Table 1). The results at least demonstrate that thermal treatment of the Mg-GOM may not have remarkably influenced the amount of Mg crosslinking between stacked GO nanosheets.

However, this thermal reduction could have led to the recovery of graphitic *sp*^2^ bonding structures and the removal of oxygen functional groups from stacked GO layers. The C/O ratio of the GOM series rose from 0.79 (pristine GOM) to 1.46 (Mg-rGOM(TH)) after thermal reduction, as shown in Table 1. Corresponding to this, the atomic content ratios obtained based on the three deconvoluted peaks from the XPS spectra were changed; when the Mg-GOM was thermally reduced, the atomic ratio of the C=C bonds increased while those of the C–O and O–C=O bonds decreased (Figure 4B). Moreover, in the FT-IR spectrum of the Mg-rGOM(TH), decreased intensities were observed for the peaks related to oxygen functional groups, such as OH stretching, C–O vibration, and alkoxy C–O stretching (Figure 4D). These results clearly demonstrate that some of the oxygen functional groups were eliminated from the GOMs.

In addition, the removal of oxygen functional groups by thermal reduction could have consequently changed the interactions between Mg^2+^ ions and GO nanosheets, which are classified according to the sites where GO nanosheets bind with Mg^2+^ ions, such as at epoxide groups, carboxylic acid groups, and graphitic basal planes. From a careful analysis of the deconvoluted C 1*s* XPS spectra of the Mg-rGOM(TH), it was found that the C–O and O–C=O peaks were repositioned at 286.58 eV and 288.38 eV, respectively. The position shifts could be attributed to an increase in the number of noncovalent interactions between the metal cations and the graphitic basal plane. This is also supported by the differences in shape and intensity of the C=C peak between the FT-IR spectra of the GOM series. The decreased intensity and broadened shoulder peak for the C=C bond (1581 cm^−1^) could validate cation binding on the π electron-rich graphitic regions of GO nanosheets [25].

### 3.4. Surface Characteristics of GOM Series

As a result of the chemical structure variations of the GOMs, some critical properties of their surfaces, such as hydrophilicity and surface charge, changed. Figure 5A displays the water contact angle of each GOM. Analogous to previous studies [32,35], the pristine GOM had a highly hydrophilic surface because of the abundance of hydrophilic oxygen functional groups attached to the surface of the GOM. The Mg-GOM exhibited a comparable contact angle value, indicating that the influence of Mg^2+^ ion crosslinking on the surface hydrophilicity was negligible. On the other hand, the Mg-rGOM(TH) possessed a more hydrophobic surface (i.e., higher contact angle) than the pristine GOM and Mg-GOM; some of the hydrophilic oxygen functional groups in the GOMs were detached as an effect of thermal reduction.

Figure 5B shows the result of measuring the surface charge (or zeta potential) of the GOM series. The surface charge of the GOM was measured at −50 mV, which is in accordance with previously reported values [36,37]. The high surface charge of the GOM was due to the plentiful oxygen functional groups, particularly rich ionizable species, such as the carboxylic groups [37]. The Mg^2+^ crosslinking increased the surface charge; thus, the Mg-GOM showed a more positive surface charge of –18.8 mV. This was likely caused by the partial neutralization of the GO surface by positively charged Mg^2+^ ions binding to some negatively charged functional groups [38]. Similarly, the Mg-rGOM(TH) exhibited fewer negatively charged ions than the GOM, but more negatively charged ions than the Mg-GOM. This can be explained by the removal of some oxygen functional groups during thermal reduction, as corroborated by the XPS, EDX, and FT-IR results.

### 3.5. The Nanochannel Size of the GOM Series

The effect of Mg^2+^ crosslinking and thermal reduction on the size of the nanochannels (i.e., interlayer spacing) of the GOM series was examined using XRD analysis. As shown in Figure 6, a sharp and high-intensity peak was observed at 10.72° during XRD analysis of the pristine GOM, corresponding to a nanochannel size of about 8.3 Å. The Mg-GOM exhibited a peak at a lower 2θ value of 6.48°. This indicates that Mg^2+^ crosslinking enlarged the size of the GO nanochannels to ~13.6 Å. This increase in GO nanochannel size after the intercalation of divalent cations was also observed in previous studies [25,26,27,28,29]; the GO film modified with divalent ions such as Mg^2+^ and Ca^2+^ exhibited larger interlayer spacing than the unmodified GO film [27]. In a previous study conducted by Chen et al. (2017), the GOMs immersed in an MgCl_2_ solution exhibited an interlayer spacing of about 13.6 Å, which is the same value as obtained in this study [24]. More recently, an Mn^2+^ ion-intercalated GOM showed a widened interlayer spacing of ~15.1 Å, compared with a pristine GOM [25].

Unlike for the Mg-GOM, the three peaks of Mg-rGOM(TH) were positioned at higher 2θ values, 9.04°, 10.93°, and 22.07°, corresponding to about 9.8 Å, 8.1 Å, and 4.0 Å, respectively. This positively shifted 2θ value (9.04°) of the largest peak indicates that the laminar GO layer was partially reduced by the heat treatment. In addition, the peak that newly appeared at 22.07°, which is generally observed in the XRD patterns for reduced GO, demonstrates that some stacked GO nanosheets were reduced with the recovery of the graphitic *sp*^2^ domain from the heat treatment. However, the reason for another peak newly appearing at 10.93° is unclear. We speculate two reasons for the creation of this peak: (1) the different degrees of reduction in the GO nanosheets constituting the GOM depending on the depth of the laminar GO layer, and (2) the oxidation of Mg^2+^ ions into magnesium carbonate or magnesium carbonate hydroxide.

Overall, the XRD data demonstrated that the Mg-rGOM(TH) possessed narrower GO nanochannels compared with the Mg-GOM. These results are in accordance with the aforementioned characterization results (SEM, EDX, XPS, and FT-IR).

### 3.6. The NF Performance of the GOM Series

Mg^2+^ ion crosslinking and partial reduction by heat treatment are expected to change the potential amount of mass transport across the GOMs because they lead to variations in some physicochemical properties of the GOMs; thus, the nanofiltration performance of the GOM series was evaluated using three different organic dye molecules: MO, MB, and RB. These organic dye molecules were selected on the basis of their molecular weights and charges. Detailed information of the chosen organic dye molecules is described in Table 2. Figure 7A shows the organic dye-rejection efficiencies of the GOM series. The pristine GOM exhibited rejection efficiencies of 85.4% for MO, 92.8% for MB, and 68.2% for RB. Despite RB having the largest molecular weight (or size) among the three chosen organic dyes, the lowest rejection efficiency was observed with RB. This relatively poor rejection rate of the GOM toward RB could have been caused by its neutral charge. In the case of MO, its negative charge could have generated electrostatic repulsion with the negatively charged surface of the GOM (shown in Figure 5B), resulting in a higher rejection efficiency than RB, despite a smaller molecular weight [39,40]. Paradoxically, even for the positively charged MB, which has the smallest molecular weight, the GOM had a higher rejection rate than that for other selected organic dyes. This might have been due to the contribution of the adsorptive removal of numerous positively charged MB molecules by the negatively charged GO laminates during separation. According to a previous study, GO was able to adsorb more MB molecules than RB molecules [41]. This result suggests that electrostatic repulsion and adsorption based on charge could play an important role in the separation mechanism of the GOM, in addition to size exclusion.

However, the Mg^2+^ ion crosslinking changed the separation performance of the GOM. As shown in Figure 7A, overall, the Mg-GOM recorded decreased rejection efficiencies (i.e., 82.8% for MO, 73.8% for MB, and 57.2% for RB) compared with the GOM. The decline in rejection efficiencies can be attributed to the enlarged GO nanochannel size caused by Mg^2+^ crosslinking. Moreover, the positively shifted surface charge from Mg^2+^ ion crosslinking could weaken the electrostatic repulsion and adsorption capacity of the Mg-GOM. In contrast to the Mg-GOM, Mg-rGOM(TH) showed higher rejection efficiencies of 95.8% for MO, 98.88% for MB, and 86.4% for RB. This can be explained by the better size exclusion enabled by the narrower GO nanochannels created after thermal reduction. Additionally, the decreased hydrophilicity of the laminar GO layer of the Mg-GOM after thermal reduction might have prevented the separation performance from deteriorating with an enlargement of GO nanochannels due to swelling in an aqueous solution [19]; therefore, these imply that filtering by size exclusion could be dominant in comparison electrostatic repulsion and adsorption in the Mg-rGOM.

Next, the Mg^2+^ ion crosslinking and thermal reduction decreased the water permeability of the GOMs, as shown in Figure 7B. Despite the bigger nanochannel size of the Mg-GOM, a lower water permeability was achieved with the Mg-GOM (2.25 L/m^2^/h/bar) than with the pristine GOM (5.26 L/m^2^/h/bar). This was because the hydration shells of the Mg^2+^ ions hindered water transport. In an aqueous environment, the GO nanochannels are filled with water molecules. In such an environment, the cations intercalated between GO laminates are hydrated, and then cations form hydration shells that surround them. These hydration shells can restrain the transport of water molecules in the GO nanochannels by attracting them instead. This phenomenon was demonstrated in a previous study [25]. Although Mn^2+^ ion-intercalated GO laminates possess bigger interlayer spacing, Mn^2+^ ion-intercalated GO laminates achieve less water permeance than the pristine GO laminates because of the hydration shells surrounding Mn^2+^ ions draw in water molecules. In addition, cations that can create bigger hydration shells with their larger charge densities tend to bind more water molecules and retain water molecules in the GO nanochannels longer.

When it comes to the Mg-rGOM(TH), in addition to the impediment of hydrated cations, compacted nanochannels could have been a significant reason for the decreased water permeability (1.95 L/m^2^/h/bar).

In comparison to other GO-based NF membranes reported in previous studies (Table 3), the water permeability of the Mg-rGOM(TH) was considerably low, but the MB rejection efficiency of the Mg-rGOM(TH) was comparable to or even higher than recently developed other GO-based NF membranes. This demonstrates that the combination of Mg^2+^ crosslinking and partial reduction could be an effective approach to enhance the rejection efficiency. In addition, Mg^2+^ crosslinking and partial reduction could have merit over other GO-based NF membranes in terms of simplicity and cost-effectiveness.

### 3.7. The Stability of the GOM Series

An evaluation of stability was carried out for the GOM series. Usually, pristine GO membranes swell easily and are damaged in aqueous conditions because of their high hydrophilicity, whereby they become physically fragile. Therefore, Mg intercalation and GO reduction were performed to reduce the vulnerability of GOMs to swelling. It can be expected that the membranes would have more resistance to swelling because of the cation–π interaction between the GO layers and the Mg cations, as well as the decrease in hydrophilicity, thereby enhancing stability in aqueous conditions. To confirm the physical durability of the membranes, a stability test was carried out in water by inducing ultrasonic power for 90 min. As shown in Figure 8, GOM and Mg-GOM were confirmed to be easily damaged by sonication, but the Mg-rGOM(TH) maintained its structure in most areas, including the central portion, for 90 min because of its improved physical stability. In addition, after 30 min sonication, the MO rejection efficiencies of the GOM and Mg-GOM(TH) were compared. As shown in Appendix A, the laminar GO layer of the 30 min sonicated pristine GOM was fully detached from the nylon support layer during the filtration test. On the other hand, the laminar GO layer of the 30 min sonicated Mg-rGOM(TH) seemed to be only partially damaged. Although the MO rejection efficiency of the 30 min sonicated Mg-rGOM(TH) decreased, its laminar GO layer could still reject the organic dye solutes. There are two possible reasons for the improved stability of the graphene layer: (1) penetration of water molecules became difficult because the hydrophilicity was decreased by the reduction treatment, and (2) the membrane structure was kept tight by crosslinking between the graphene nanosheets and Mg cations. Therefore, it is shown that the structural durability of nanomaterial-based membranes could be improved by cation intercalation and oxidation control. This approach is expected to improve the possibility of using nanomaterials as nanofiltration membranes.

## 4. Conclusions

In this study, we modified GOMs using Mg^2+^ ion crosslinking and partial reduction in simple two steps, i.e., dipping and heat treatment, to obtain robust, high-performance GO-based NF membranes. Conclusively, these two facile modification steps secured more compact and tightly interlocked GO nanochannels due to their cation–π interactions and partially eliminated oxygen functional groups. This significantly enhanced the GOM’s rejection efficiencies for organic dyes with a molecular weight between 300 and 500 g/mol, regardless of their charge, and furthermore remarkably improved the stability of the GOM in an aqueous environment. However, the hydration shells that formed around the Mg^2+^ ions inserted into the GO nanochannels because of the high charge density of the Mg^2+^ ions impeded the transport of water molecules through the GO nanochannels, resulting in decreased water permeability; further studies are required to address this issue. Despite the decreased water permeability, combining Mg^2+^ ion crosslinking and partial reduction is a feasible strategy for developing high-performance NF membranes that can precisely sieve small molecules within a specific size range because of their simplicity and scalability.

## Figures and Tables

**Figure 1 membranes-12-00402-f001:**
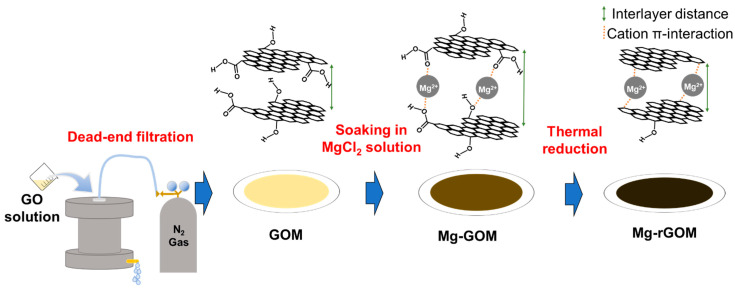
Illustration of the fabrication of the GOM series.

**Figure 2 membranes-12-00402-f002:**
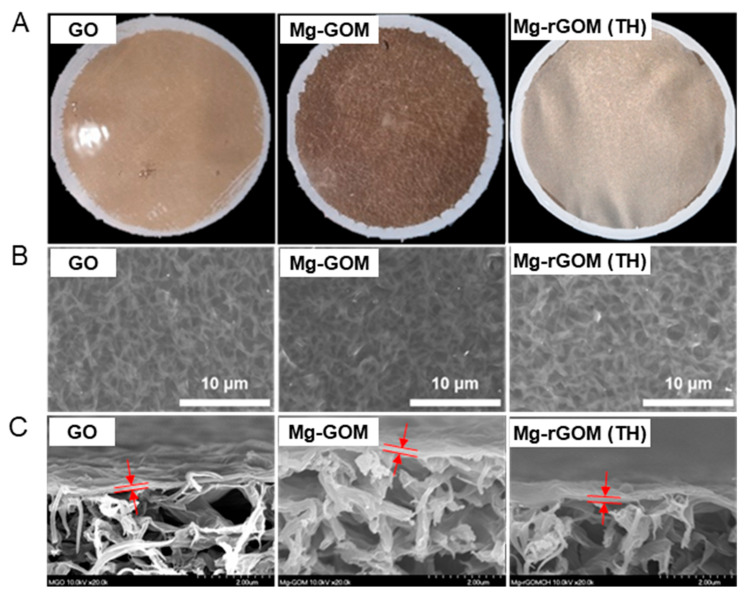
(**A**) Photographs, (**B**) top-view SEM images, and (**C**) cross-sectional SEM images of GOM, Mg-GOM, and Mg-rGOM(TH).

**Figure 3 membranes-12-00402-f003:**
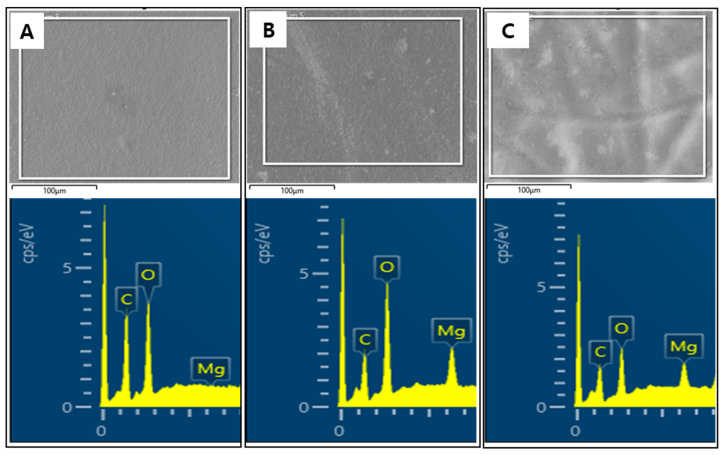
EDX results of (**A**) GOM, (**B**) Mg-GOM, and (**C**) Mg-rGOM(TH).

**Figure 4 membranes-12-00402-f004:**
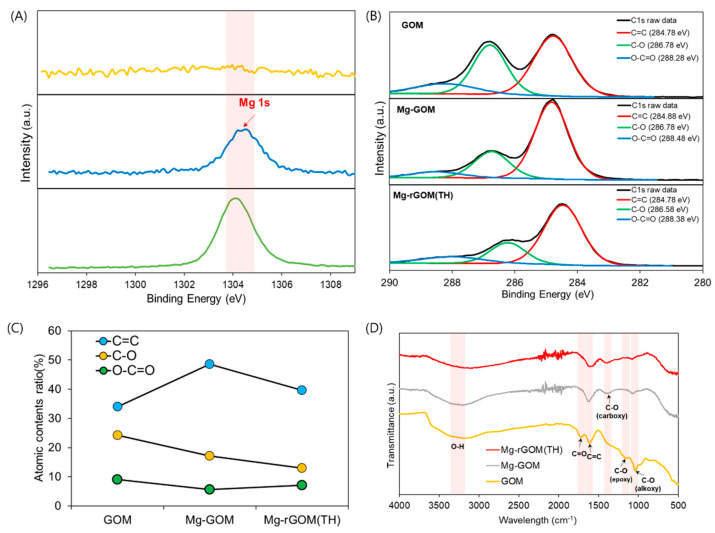
(**A**) The high-resolution XPS spectra for Mg 1*s*, (**B**) the high-resolution deconvoluted XPS spectra for C 1*s*, (**C**) the atomic content ratio of carbon-related bonds (C=C, C–O, and O–C=O), and (**D**) the FT-IR spectra of the GOM series.

**Figure 5 membranes-12-00402-f005:**
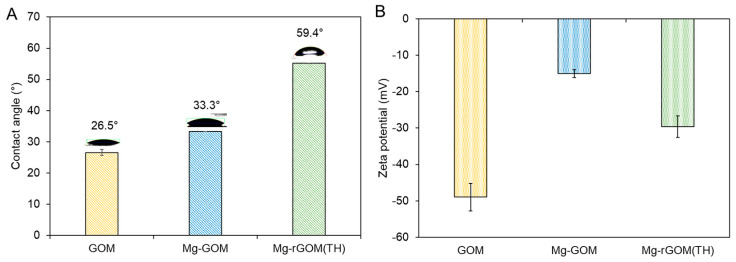
(**A**) Water contact angle and (**B**) surface charge of GOM series.

**Figure 6 membranes-12-00402-f006:**
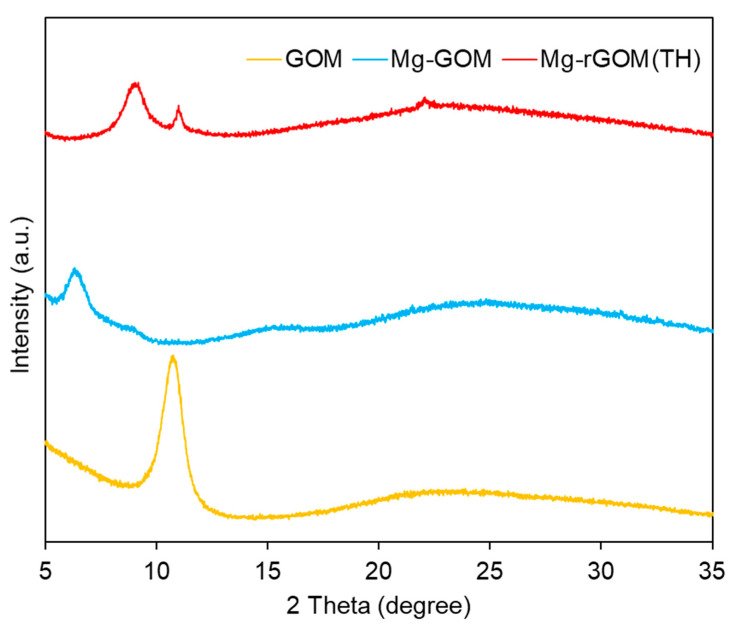
XRD patterns of the GOM series.

**Figure 7 membranes-12-00402-f007:**
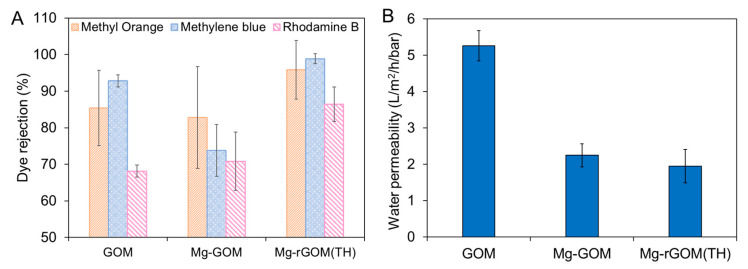
(**A**) Dye rejection rates and (**B**) pure water permeability of the GOM series.

**Figure 8 membranes-12-00402-f008:**
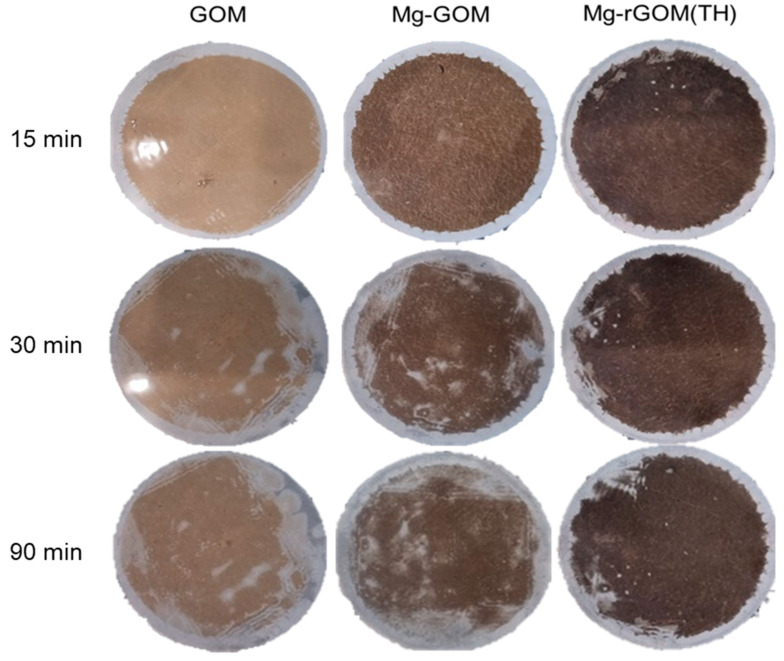
Structural durability of GOM, Mg-GOM, and Mg-rGOM(TH) membranes, observed at different sonication times in the water.

**Table 1 membranes-12-00402-t001:** Atomic composition of the GOM series obtained from EDX results.

Membrane	Atomic Composition (wt.%)			C/O Ratio
	C	O	Mg	
GOM	48.61	61.18	0.20	0.79
Mg-GOM	38.16	39.54	6.77	0.97
Mg-rGOM(TH)	46.24	31.75	7.15	1.46

**Table 2 membranes-12-00402-t002:** Organic dye molecules used for evaluating nanofiltration performance of the GOM series [42].

Organic Dye	Molecular Weight (g/mol)	Size (Å^3^)	Charge
Methyl orange (MO)	327.34	17.93 × 7.54 × 6.02	Negative
Methylene blue (MB)	319.85	16.94 × 8.24 × 4.55	Positive
Rhodamine B (RB)	479.02	18.54 × 14.35 × 9.14	Neutral

**Table 3 membranes-12-00402-t003:** Organic dye molecules used for evaluating nanofiltration performance of the GOM series.

GOM	Permeability (L/m^2^/h/bar)	Organic Dye Rejection (%)			Ref.
		MO	MB	RB	
Mg-rGOM(TH)	1.95	95.8	98.9	86.4	This study
GO/SiO_2_	44.2	91.0			[43]
GO/Fe_3_O_4_	296			98.0	[44]
GO/NH_2_-Fe_3_O_4_	15.6		70.0		[45]
GO/glycine/g-C_3_N_4_	207		87.0		[46]
GO/TiO_2_ nanosheet	9.36	97.3	98.8		[47]
GO/MB	3.83	96.37			[48]
Reduced preoxidized GO	5.3	97.5			[49]
GO/isophorone diisocyanate	80–100	97	97.7	96.2	[50]
GO/hydroxylated graphene	<24.4	<99.7		<99.7	[51]
GO/Ag	20.8–33.9		94.6–96.8	77.9–84.2	[52]

## Data Availability

Not applicable.

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
