# Peer review of "Enhancing the Dye-Rejection Efficiencies and Stability of Graphene Oxide-Based Nanofiltration Membranes via Divalent Cation Intercalation and Mild Reduction"

_membranes, 2022, doi:10.3390/membranes12040402_

Round 1
Reviewer 1 Report
This paper focused on the development of a 2D GO membrane for organic dye nanofiltration. The authors mentioned that 2D GO membranes have advanced performances, but it has limitation in physical and chemical stability. To solve the problem, the 2D GO membrane was crosslinked with divalent ions. Those membranes have different surface properties and different spaces. From the unique membrane structure, the rejections of organic dyes were controlled. This scope and result are well proper to be accepted in Membranes journal. All English type is perfect. Therefore, it can provide all the readers with interests without any reading issues.
Author Response
Referee #1
Comment 1: This paper focused on the development of a 2D GO membrane for organic dye nanofiltration. The authors mentioned that 2D GO membranes have advanced performances, but it has limitation in physical and chemical stability. To solve the problem, the 2D GO membrane was crosslinked with divalent ions. Those membranes have different surface properties and different spaces. From the unique membrane structure, the rejections of organic dyes were controlled. This scope and result are well proper to be accepted in Membranes journal. All English type is perfect. Therefore, it can provide all the readers with interests without any reading issues.
Our response) We thank the referee for providing us with such a positive evaluation.
Reviewer 2 Report
In the work, flat sheet membranes based on graphene oxide have been developed and studied. Modification of membranes with magnesium salt followed by reduction resulted in a significant increase in the retention of nanometer-sized organic dyes up to 98.88%. The results of the article are new and relevant. However, there are remarks that must be taken into account before the article can be published:
- Filtration results (permeability and rejection) are not compared to the best published data. Therefore, the relevance of the work is not so obvious.
- Figure 8 shows photos of membranes after sonication. However, the photos are not informative. The results cannot be analyzed in the absence of data on the filtration of dyes through the treated membranes.
- It would be necessary to study the dimensions of the transport channels of the membranes by capillary flow porosimetry in order to interpret the results of the retention. The results of X-ray spectroscopy are indirect, since they cannot differentiate transport and dead-end pores. In addition, X-rays were carried out in air, and not in the feed liquid.
- Permeability units, in particular in Figure 7, should be generally accepted. Abbreviations like LMH are confusing and can be misinterpreted.
- In the introduction and discussion of the work, the literature on the nanofiltration separation of dyes is not well enough cited. For example, articles of such leading scientific teams as:
Oyarce, E., Santander, P., Butter, B., Pizarro, G. D. C., & Sánchez, J. (2021). Use of sodium alginate biopolymer as an extracting agent of methylene blue in the polymer‐enhanced ultrafiltration technique. Journal of Applied Polymer Science, 138(34), 50844.
Wang, Q., Ju, J., Tan, Y., Hao, L., Ma, Y. & Wu, Y. (2019). Controlled Synthesis of Sodium Alginate Electrospun Nanofiber Membranes for Multi-occasion Adsorption and Separation of Methylene Blue. Carbohydrate Polymers, 205, 125-154.
T.Anokhina, E.Dmitrieva, A.Volkov. Recovery of Model Pharmaceutical Compounds from Water and Organic Solutions with Alginate-Based Composite Membranes. Membranes, 12 №2 (2022) 235.
A.A.Yushkin, T.S.Anokhina, A.V.Volkov. Application of cellophane films as nanofiltration membranes. Petroleum Chemistry. V.55, №9, (2015), 746–752.
Sanchuan Yua, Zhiwen Chen, Qibo Cheng, Zhenhua Lü, Meihong Liu, Congjie Gao. Application of thin-film composite hollow fiber membrane to submerged nanofiltration of anionic dye aqueous solutions. Separation and Purification Technology, V. 88, 2012, P. 121-129.
Naeme Nikoo, Ehsan Saljoughi. Preparation and characterization of novel PVDF nanofiltration membranes with hydrophilic property for filtration of dye aqueous solution. Applied Surface Science. V.413, 2017, P.41-49
Author Response
Referee #2
Comment 1: In the work, flat sheet membranes based on graphene oxide have been developed and studied. Modification of membranes with magnesium salt followed by reduction resulted in a significant increase in the retention of nanometer-sized organic dyes up to 98.88%. The results of the article are new and relevant. However, there are remarks that must be taken into account before the article can be published:
Our response) We are grateful to the referee for reviewing our manuscript. We have carefully responded to and addressed the comments from the referee.
Comment 2: Filtration results (permeability and rejection) are not compared to the best published data. Therefore, the relevance of the work is not so obvious.
Our response) We are grateful to the referee for reviewing our manuscript. We have carefully responded to and addressed the comments from the referee. As the reviewer’s suggestion, we have added the performance comparison of our GO membrane to other nanofiltration membranes reported in previous studies.
Line 368-397, Page 10
As a performance comparison of the Mg-rGOM(TH) to other GO-based NF mem-branes reported in previous studies, the water permeability of the Mg-rGOM(TH) is considerably low, but the MB rejection efficiency of the Mg-rGOM(TH) is comparable or even higher than recently-developed other GO-based NF membranes. This demonstrates combination of Mg2+ crosslinking and partial reduction could be an effective approach to enhance the rejection efficiency. In addition, Mg2+ crosslinking and partial reduction could have merit over the other GO-based NF membranes in terms of sim-plicity and cost-effectiveness.
Table 3. Organic dye molecules used for evaluating nanofiltration performance of the GOM series.
|
GOM |
Permeability ( L/m2/h/bar) |
Organic dye rejection (%) |
Ref. |
||
|
MO |
MB |
RB |
|||
|
Mg-rGOM(TH) |
1.95 |
95.8 |
98.9 |
86.4 |
This study |
|
GO/SiO2 |
44.2 |
91.0 |
|
|
[43] |
|
GO/Fe3O4 |
296 |
|
|
98.0 |
[44] |
|
GO/NH2-Fe3O4 |
15.6 |
|
70.0 |
|
[45] |
|
GO/glycine/g-C3N4 |
207 |
|
87.0 |
|
[46] |
|
GO/TiO2 nanosheet |
9.36 |
97.3 |
98.8 |
|
[47] |
|
GO/MB |
3.83 |
96.37 |
|
|
[48] |
|
Reduced preoxidized GO |
5.3 |
97.5 |
|
|
[49] |
|
GO/isophorone diisocyanate |
80-100 |
97 |
97.7 |
96.2 |
[50] |
|
GO/hydroxylated graphene |
<24.4 |
<99.7 |
|
<99.7 |
[51] |
|
GO/Ag |
20.8-33.9 |
|
94.6-96.8 |
77.9-84.2 |
[52] |
- Yang, K.; Pan, T.; Hong, S.; Zhang, K.; Zhu, X.; Chen, B. Ultrathin graphene oxide membrane with constructed tent-shaped structures for efficient and tunable molecular sieving. Environ. Sci. Nano 2020, 7, 2373-2384.
- Zhang, M.; Guan, K.; Shen, J.; Liu, G.; Fan, Y.; Jin, W. Nanoparticles@ rGO membrane enabling highly enhanced water permeability and structural stability with preserved selectivity. AIChE J. 2017, 63, 5054-5063.
- Dong, L.; Li, M.; Zhang, S.; Si, X.; Bai, Y.; Zhang, C. NH2-Fe3O4-regulated graphene oxide membranes with well-defined laminar nanochannels for desalination of dye solutions. Desalination 2020, 476, 114227.
- Wu, Z.; Gao, L.; Wang, J.; Zhao, F.; Fan, L.; Hua, D.; Japip, S.; Xiao, J.; Zhang, X.; Zhou, S.-F. Preparation of glycine mediated graphene oxide/g-C3N4 lamellar membranes for nanofiltration. J. Membr. Sci. 2020, 601, 117948.
- Yu, J.; Zhang, Y.; Chen, J.; Cui, L.; Jing, W. Solvothermal-induced assembly of 2D-2D rGO-TiO2 nanocomposite for the construction of nanochannel membrane. J. Membr. Sci. 2020, 600, 117870.
- Hou, J.; Chen, Y.; Shi, W.; Bao, C.; Hu, X. Graphene oxide/methylene blue composite membrane for dyes separation: Formation mechanism and separation performance. Appl. Surf. Sci. 2020, 505, 144145.
- Chang, Y.; Shen, Y.; Kong, D.; Ning, J.; Xiao, Z.; Liang, J.; Zhi, L. Fabrication of the reduced preoxidized graphene-based nanofiltration membranes with tunable porosity and good performance. RSC Adv. 2017, 7, 2544-2549.
- Zhang, P.; Gong, J.-L.; Zeng, G.-M.; Deng, C.-H.; Yang, H.-C.; Liu, H.-Y.; Huan, S.-Y. Cross-linking to prepare composite graphene oxide-framework membranes with high-flux for dyes and heavy metal ions removal. Chem. Eng. J. 2017, 322, 657-666.
- Deng, H.; Huang, J.; Qin, C.; Xu, T.; Ni, H.; Ye, P. Preparation of high-performance nanocomposite membranes with hydroxylated graphene and graphene oxide. J. Water Process Eng. 2021, 40, 101945, doi:https://doi.org/10.1016/j.jwpe.2021.101945.
- Yang, K.; Huang, L.-j.; Wang, Y.-x.; Du, Y.-c.; Zhang, Z.-j.; Wang, Y.; Kipper, M.J.; Belfiore, L.A.; Tang, J.-g. Graphene oxide nanofiltration membranes containing silver nanoparticles: Tuning separation efficiency via nanoparticle size. Nanomaterials 2020, 10, 454.
Comment 3: Figure 8 shows photos of membranes after sonication. However, the photos are not informative. The results cannot be analyzed in the absence of data on the filtration of dyes through the treated membranes.
Our response) We agree with what the reviewer pointed out, and we appreciate the comment. After 30-min sonication, the laminar GO layer of the pristine GOM was fully detached from the nylon support layer. On the other hand, the laminar GO layer of the Mg-rGOM seems to be partially damaged by 30-min sonication. The methyl orange dye rejection efficiency of the Mg-rGOM after 30-min sonication decreased from 82.2% to 56.4%. This indicates cation crosslinking and partially reduction can enhance the physical durability of the laminar GO layer. We added the result into Supplementary Information as Figure S2.
Line 416-423, Page 11-12
In addition, after 30-min sonication, the MO rejection efficiencies of the GOM and Mg-GOM(TH) were compared. As shown in Figure S2, the laminar GO layer of the 30-min sonicated pristine GOM was fully detached from the nylon support layer during the filtration test. On the other hand, the laminar GO layer of the 30-min sonicated Mg-rGOM(TH) seems to be only partially damaged. Although the MO rejection efficiency of the 30-min sonicated Mg-rGOM(TH) decreased, its laminar GO layer could still reject the organic dye solutes.
Comment 4: It would be necessary to study the dimensions of the transport channels of the membranes by capillary flow porosimetry in order to interpret the results of the retention. The results of X-ray spectroscopy are indirect, since they cannot differentiate transport and dead-end pores. In addition, X-rays were carried out in air, and not in the feed liquid.
Our response) Thank you to the reviewer for highlighting this part. We agree that the capillary flow porosimetry analysis is worthy of investigating the dimensions of the transport channels of membranes. However, we believe that porosimetry is not appropriate to the pore structure characterization of our GO membranes. Our GO membrane consists of an ultra-thin laminar GO layer and nylon support layer. The thickness of the laminar GO layer, which acts as a selective layer, is extremely smaller (about 150 nm) than the nylon support layer. Even if we obtain any results from the analyses of the GO membrane using porosimetry experiments, the results would be inaccurate due to the effect of the thick and porous nylon support layer. In addition, the nanochannel size of GO membranes is in the range of nanofiltration. Porosimetry analysis is generally used for pore structure characterization of ultrafiltration and microfiltration membrane and support layer of thin-film composite membranes; thus, due to the ultra-thin thickness and sub-nanometer nanochannels size of the GO membranes, the pore structure characterization using porosimetry could be not suitable for the nanochannel characterization of GO membranes.
Also, XRD is a generally accepted approach for measuring the nanochannel size of GO membranes [1, 2]. Nevertheless, as the reviewer pointed out, XRD analysis in wet-condition GO membrane should be provided. Unfortunately, we were not able to conduct the analysis of the wet-condition GO membrane using XRD due to an issue of equipment failure. We seek the kind understanding of the reviewer on this matter.
Comment 5: Permeability units, in particular in Figure 7, should be generally accepted. Abbreviations like LMH are confusing and can be misinterpreted.
Our response) We are grateful to the referee for reviewing our manuscript. We have changed the unit of water permeability from LMH into L/m2/h.
Comment 6: In the introduction and discussion of the work, the literature on the nanofiltration separation of dyes is not well enough cited. For example, articles of such leading scientific teams as:
Oyarce, E., Santander, P., Butter, B., Pizarro, G. D. C., & Sánchez, J. (2021). Use of sodium alginate biopolymer as an extracting agent of methylene blue in the polymer‐enhanced ultrafiltration technique. Journal of Applied Polymer Science, 138(34), 50844.
Wang, Q., Ju, J., Tan, Y., Hao, L., Ma, Y. & Wu, Y. (2019). Controlled Synthesis of Sodium Alginate Electrospun Nanofiber Membranes for Multi-occasion Adsorption and Separation of Methylene Blue. Carbohydrate Polymers, 205, 125-154.
T.Anokhina, E.Dmitrieva, A.Volkov. Recovery of Model Pharmaceutical Compounds from Water and Organic Solutions with Alginate-Based Composite Membranes. Membranes, 12 №2 (2022) 235.
A.A.Yushkin, T.S.Anokhina, A.V.Volkov. Application of cellophane films as nanofiltration membranes. Petroleum Chemistry. V.55, №9, (2015), 746–752.
Sanchuan Yua, Zhiwen Chen, Qibo Cheng, Zhenhua Lü, Meihong Liu, Congjie Gao. Application of thin-film composite hollow fiber membrane to submerged nanofiltration of anionic dye aqueous solutions. Separation and Purification Technology, V. 88, 2012, P. 121-129.
Naeme Nikoo, Ehsan Saljoughi. Preparation and characterization of novel PVDF nanofiltration membranes with hydrophilic property for filtration of dye aqueous solution. Applied Surface Science. V.413, 2017, P.41-49
Our response) we have added some of the references that the reviewer suggested in our revised manuscript.
Line 37-40, Page 1
Among these processes for securing water from alternative sources, nanofiltration (NF) has increasingly been employed because it has several attractive merits, such as the capability to reject ions up to the divalent level (e.g., heavy metal ions), the ability to remove the wid-est range of organic contaminants, and its relatively low energy consumption [5-10].
- Nikooe, N.; Saljoughi, E. Preparation and characterization of novel PVDF nanofiltration membranes with hydrophilic property for filtration of dye aqueous solution. Applied Surface Science 2017, 413, 41-49, doi:https://doi.org/10.1016/j.apsusc.2017.04.029.
- Yu, S.; Chen, Z.; Cheng, Q.; Lü, Z.; Liu, M.; Gao, C. Application of thin-film composite hollow fiber membrane to submerged nanofiltration of anionic dye aqueous solutions. Separation and Purification Technology 2012, 88, 121-129, doi:https://doi.org/10.1016/j.seppur.2011.12.024.
- Yushkin, A.A.; Anokhina, T.S.; Volkov, A.V. Application of cellophane films as nanofiltration membranes. Petroleum Chemistry 2015, 55, 746-752, doi:10.1134/S0965544115050114.
- Anokhina, T.; Dmitrieva, E.; Volkov, A. Recovery of Model Pharmaceutical Compounds from Water and Organic Solutions with Alginate-Based Composite Membranes. Membranes 2022, 12, 235.
- END -
[1] Guo J, Bao H, Zhang Y, Shen X, Kim J-K, Ma J, et al. Unravelling intercalation-regulated nanoconfinement for durably ultrafast sieving graphene oxide membranes. Journal of Membrane Science. 2021;619:118791.
[2] Du Y-c, Huang L-j, Wang Y-x, Yang K, Zhang Z-j, Wang Y, et al. Preparation of graphene oxide/silica hybrid composite membranes and performance studies in water treatment. Journal of Materials Science. 2020;55:11188-202.
Round 2
Reviewer 2 Report
The authors revised the work well. The article can be published in the Membranes.